# Mask Adherence and Social Distancing in Houston, TX from January to April 2021

**DOI:** 10.3390/ijerph20032723

**Published:** 2023-02-03

**Authors:** Simone Trevas, Kathleen Manuel, Raja Malkani, Deanna Hoelscher

**Affiliations:** 1Epidemiology, University of Texas Health Science Center at Houston, Houston, TX 77030, USA; 2Michael & Susan Dell Center for Healthy Living, University of Texas Health Science Center at Houston, Austin, TX 78701, USA; 3Michael & Susan Dell Center for Healthy Living, University of Texas Health Science Center at Houston, Campus Dean, Austin, TX 78701, USA

**Keywords:** mask-wearing, social distancing, COVID-19, observational study, outdoors

## Abstract

Shortly after the identification of COVID-19, public health experts recommended the use of face masks and social distancing to slow the spread of the virus. Early research indicates that there are associations between gender, age, and mask-wearing behavior. The primary aim of this paper was to explore how demographics, location, and mask mandates may affect COVID-19 mitigation strategies such as mask-wearing and social distancing. A prospective, cross-sectional observational study was conducted in Houston, TX from January to April 2021 at three outdoor locations: an urban park, an urban park with a trail, and a farmer’s market. During each two-hour data collection period, trained observers recorded the total number of people in the designated spaces; people were categorized by approximate age, sex, race/ethnicity, physical activity level, social distancing, and mask adherence using the Systematic Observation of Mask Adherence and Distancing (SOMAD) protocol. Multivariable logistic regression was used to determine associations with gender, race, age, location, and the mask mandate. A total of 7778 observations were recorded after exclusion of inconclusive demographic data. Females had higher odds, reported as an odds ratio, of mask use than males (OR = 1.35, 95% CI 1.18–1.54). Compared to White individuals, Asian individuals had higher odds of mask use (OR = 1.84, 95% CI 1.48–2.30). The odds of mask use were higher while the Texas mask mandate was in effect (OR = 1.60, 95% CI 1.40–1.84). Regarding location, the odds of mask use were much higher in the urban park than in the urban park with a trail (OR = 13.33). Individuals had higher odds of social distancing at the urban park with a trail compared to the farmer’s market (OR = 4.61, 95% CI 4.10–5.17). Mask wearing and social distancing behaviors differ by demographics, locality, and mask mandate. Thus, state policies can be effective tools to encourage mask wearing for disease mitigation.

## 1. Introduction

Shortly after identifying COVID-19, public health experts recommended the use of face masks to slow the spread of the virus, since it is spread by respiratory droplets [1]. Because of mixed messages about the effectiveness of wearing masks, adoption of this safety measure has been piecemeal [2]; however, recent research has shown more positive effects of mask-wearing and virus spread [3]. Epidemiologic data have also recently confirmed the relationship between masking mandates and reduced COVID-19 community case incidence [3].

Early research indicates that there are associations between gender, age, and mask-wearing behavior [4]. A study conducted by Haischer et al. revealed that the odds of a female wearing a mask were greater than that of males (aOR, “adjusted odds ratio” = 1.47) and that the odds of older adults (>65 years old) wearing a mask were greater than middle and young age groups (aOR = 3.43) [4]. Another study conducted by Khubchandani et al. found that women in older age groups (≥36 years) were more likely to wear masks than their male and younger (≤35 years) counterparts [5]. With regard to the relationship between mask-wearing behavior and race, Hearne et al. found that compared with White participants, Black (OR = 2.24), Latina/Latino (OR = 1.62), and Asian (OR = 2.87) participants had higher odds of wearing masks during the COVID-19 pandemic [6].

Since COVID-19 is still a relatively new disease, there is little published research about the relations between demographic characteristics and mitigation strategies. In addition, few studies have shown differences in these mitigation strategies in terms of locale, and there is little evidence that there are differences in mask-wearing in response to a mask mandate or lack thereof [7]. Thus, the primary aim of this paper was to explore how location-specific contexts and mask mandates are associated with COVID-19 mitigation strategies such as mask-wearing and social distancing. Data from this study could potentially inform policy implementation for future pandemics.

## 2. Materials and Methods

This was a prospective, serial, cross-sectional observational study. All procedures involved public observation and received an exemption from the Institutional Review Board at the University of Texas Health Science Center at Houston. For the safety of the data collectors, only outdoor public spaces were observed. In addition, all data collectors were required to wear a face mask and maintain at least six feet from other people in the observation setting [7].

Data collectors were trained in the Systematic Observation of Mask Adherence and Distancing (SOMAD) protocol that builds on similar direct observation measurements such as SOFIT and SOPLAY [8,9]. In the SOMAD protocol, people are observed at a distance and there is no direct contact with the persons being observed [7]. During each two-hour data collection period, trained observers recorded the total number of people in the designated spaces; people were categorized by approximate age, sex, race/ethnicity, physical activity level, social distancing, and mask adherence. All observations took place at outdoor public locations in Houston, Texas, which included: (1) an urban park; (2) an urban park with a trail; and (3) a farmer’s market. Both parks are large public parks that include picnicking and recreational areas. During the COVID-19 pandemic, the observed farmer’s market remained open on Saturdays and included various vendors [10].

Observations were conducted from 10 a.m. to 12 p.m. on Saturdays for the farmer’s market, and on Wednesdays and Fridays between the hours of 11 a.m. and 2:40 p.m. at the parks. To record and submit data, data collectors used their cell phones and the standard Google Form entry developed by Cohen et al. (2020) [7] was used.

### 2.1. Mandated Policy Efforts to Mitigate COVID-19

On 2 July 2020, Texas Governor Greg Abbott mandated mask-wearing for Texans in counties with at least 20 COVID-19 cases. [11]. This mandate applied to people in situations when social distancing was not possible, when people were in an indoor business or publicly accessible building, and when people were outdoors in public [11]. On 3 August 2020, the Mayor of Houston, Sylvester Turner, directed the Houston Police Department to issue USD 250 fines to anyone not wearing a mask or face covering required by the state mandate [12]. On 10 March 2021, Governor Abbott lifted the mask mandate in Texas, and all businesses were allowed to reopen at 100% capacity [13].

### 2.2. Dependent Variables

The dependent variables for this study were mask-wearing and social distancing behavior from 20 January to 30 April 2021. Mask-wearing was defined as an individual wearing a mask that completely covered both their mouth and nose. Social distancing was defined as being six feet away from other people. Responses from the Google Form were coded so that 1 = “Yes” and 0 = “No”.

### 2.3. Independent Variables

The independent demographic variables included: race, which includes White (referent), Black, Latino, and Asian; gender measured by 1 = “Female” 0 = “Male”, with male as the referent; age, which was categorized as toddlers: 0–2 years old, children: 3–12 years old, teens: 13–19 years old, adults: 20–59 years old (referent), and seniors: 60+ years old. Other variables included implementation of the mask mandate, which was defined as 1 before and until 9 March when the mask mandate was still in place, and 0 after 9 March (referent) when the mask mandate was lifted. Locations included an urban park, an urban park with a trail, and a local farmer’s market.

### 2.4. Analysis

Descriptive statistics for gender, race, age, and location were downloaded from Google Sheets for analysis. Multivariable logistic regressions were conducted using STATA/IC (version 16.1, Houston, TX and Austin, TX, USA) to determine the association between gender, race, age, location, and the mask mandate on mask-wearing and social distancing behaviors. Observations were divided into two categories: mask mandate in place (20 January to 9 March) and mask mandate lifted (10 March to 30 April) in order to test for significant differences between the two time periods. Two separate logistic regression models were run: one for mask-wearing behavior and one for social distancing behavior. Both logistic regression models presented coefficients in odds ratios along with *p*-values and 95% confidence intervals.

## 3. Results

A total of 8002 observations were made in Houston, Texas between 20 January and 30 April 2021 (Table 1) with people of unknown race, age group, and nonbinary gender excluded from the analysis (*n* = 224), leaving a total of 7778 observations. Out of the 7778 observations, 62.7% of individuals were White, 11.2% were Black, 16.4% were Latino, and 9.7% were Asian. Most (53.4%) of the individuals observed were female, and the age distribution was as follows: 0.50% of individuals were toddlers (0–2 years old), 6.2% were children (3–12 years old), 1.2% were teens (13–19 years old), 80.3% were adults (20–59 years old), and 11.9% were seniors (60+ years old). In total, 44.3% of people wore a mask from 20 January to 30 April 2021.

On average, 83.8% of people wore masks at the farmer’s market, 30.4% of people wore masks at the city park, and 3.8% of people wore masks at the neighborhood park with a trail. With regard to the mask mandate, on average, 87.0%, 35.7%, and 5.1% of people wore masks at a farmer’s market, urban park, and an urban park with a trail, respectively, during the mandate, while 81.4%, 26.6%, and 2.6% of people wore masks at those sites when the mandate was lifted, respectively (Table 2).

Table 3 includes the logistic regression estimates for differences in mask-wearing behavior by gender, race, age, location, and when the mask mandate was in effect versus when it was lifted. After controlling for confounding variables, the results indicated that gender, race, age, location, and the mask mandate were all significantly associated with mask-wearing behavior from late-January to late-April 2021.

Females had higher odds of mask use than males (OR = 1.35, 95% CI 1.18–1.54). Compared to White individuals, Asian individuals had higher odds of mask use (OR = 1.84, 95% CI 1.48–2.30), whereas mask use for Black and Latino individuals did not significantly differ from White individuals, as shown in Table 3.

The odds of mask use were higher while the mask mandate was in effect (OR = 1.60, 95% CI 1.40–1.84). The odds of mask use were much higher in the urban park than in the urban park with a trail (OR = 13.33).

Compared to adults, the odds of mask use were lower among toddlers (OR = 0.01, 95% CI 0.00–0.10) and children (OR = 0.24, 95% CI 0.19–0.31), but higher among teenagers (OR = 1.54, 95% CI 0.91–2.61) and seniors (OR = 1.54, 95% CI 1.24–1.91).

Finally, Table 4 includes the logistic regression estimates for differences in social distancing behavior by gender, race, age, location, and mask mandate. Females had lower odds of social distancing compared to males (OR = 0.66, 95% CI 0.59–0.73). Latino and Asian individuals had lower odds of social distancing compared to White individuals (OR = 0.75, 95% CI 0.65–0.87) and (OR = 0.68, 95% CI 0.56–0.82), respectively, whereas Black individuals had the highest odds of social distancing (OR = 1.19, 95% CI 1.01–1.40). The odds of social distancing were higher when the mask mandate was in effect (OR = 1.21, 95% CI 1.09–1.34). Compared to adults, children and teenagers had lower odds of social distancing, whereas seniors had the highest odds of social distancing. Finally, individuals had higher odds of social distancing at the urban park with a trail compared to the farmer’s market (OR = 4.61, 95% CI 4.10–5.17).

## 4. Discussion

There were significant differences in mask-wearing behavior by gender, racial/ethnic group, location, and/or during the mask mandate in Texas. Evidence from previous studies suggests that wearing a mask and distancing, or staying at least six feet apart from others, are effective strategies for preventing the spread of COVID-19 [5,14]. Despite this knowledge, there have been mixed messages about the effectiveness of these preventive measures, hence they were not universally adopted in the U.S. during the pandemic [15]. Identifying the demographic characteristics of people who do and do not comply with mask-wearing and social distancing measures as well as mask-wearing at locations they frequent is valuable knowledge for public health professionals and decision-makers regarding policy and messaging intervention for future pandemics.

The odds of females wearing a mask were higher than that of males after controlling for confounders. One potential explanation for this finding is that males are more inclined to take risks than females and thus, might not wear masks as often [16]. The tendency for females to wear masks more than males is consistent with findings in other research studies. For instance, a study conducted by Haischer et al. found that the odds of a female wearing a mask were greater than those of males (aOR = 1.47) [4], which was slightly greater than the odds ratio reported in this study (1.35).

Asian individuals had higher odds of mask use compared to White individuals, whereas the odds of mask use for Black and Latino individuals did not differ significantly from White individuals. These results differ from what was found in the literature; for example, a report by Hearne et al. found that compared with White participants, Black (OR = 2.24), Latina or Latino (OR = 1.62), and Asian (OR = 2.87) participants had higher odds of wearing masks during the COVID-19 pandemic [6]. One potential reason for this is because the study by Hearne et al. was a cross-sectional, self-reported national survey, whereas our study included direct observations [6].

After controlling for gender and race, the odds of mask-wearing were higher before the mask mandate was lifted in Texas. This emphasizes the role of mandates in encouraging public health-related behaviors [17].

With regard to the second research question, there was a significant difference in mask-wearing and/or social distancing behavior between the urban park with a trail, the urban park, and the farmer’s market. The odds of individuals wearing a mask at the urban park were about 13 times as high as at the urban park with the trail. A plausible explanation for this observation is that the urban park is surrounded by the Texas Medical Center, which could make people more cognizant of their mask-wearing behavior. In addition, since some of the people frequenting the park may work in the health care industry, they may be more inclined to wear a mask. Compared to the farmer’s market, the odds of mask use was lower at the urban park (OR = 0.08, 95% CI 0.07–0.09) and the urban park with a trail (OR = 0.006, 95% CI 0.005–0.008). The high odds of mask use at the farmer’s market is not surprising because of the signage at the front entrance that said, “No mask, no entry” and there were also several police officers that walked around the farmer’s market and observed people.

Individuals had higher odds of social distancing at the urban park with a trail compared to the farmer’s market (OR = 4.61, 95% CI 4.10–5.17), whereas individuals had much lower odds of wearing a mask at the urban park with a trail. A possible explanation for this could be that individuals felt safe not wearing a mask when they were at least six feet away from other people. Alternatively, people using this park often use a running trail, and for people engaged in vigorous physical activity such as running, it can be uncomfortable to wear a mask [18].

Regarding the third and final research question, there were significant differences in mask-wearing behaviors by age group. Compared to adults, toddlers and children had lower odds of mask use, whereas teenagers and seniors had higher odds of mask use. These results are somewhat consistent with other studies; specifically, Haischer et al. found that the odds of older adults (>65 years old) wearing a mask were greater than the middle and young age groups (aOR = 3.43) [4]. One possible explanation for adults and seniors wearing masks more than children is that while the mask mandate was in place in Texas, children under 10 years old were exempt from wearing a mask [19]. Another possible explanation is that older adults are more likely to become infected with COVID-19 and tend to have worse symptoms than the younger population, which may increase the possibility of them taking preventive measures such as wearing a mask [20].

### Limitations and Strengths

One of the limitations of this study is that due to its observational nature, the data collection process might not be as accurate for certain variables such as race, since it is difficult to accurately assess someone’s race or ethnicity merely by observing them. Furthermore, age misclassification could be present since certain races tend to look younger than others [21]. Because the locations were limited to outdoor spaces, the weather affected the ability to collect data on several occasions. In addition, individuals may have not been as compliant with mask-wearing behaviors as they are indoors because data suggest that the risk of contracting COVID-19 outdoors is lower than contracting the virus indoors [22]. In addition, sampling bias may be present because those who left their homes during the pandemic may be healthier and vaccinated compared with those who chose to stay home during this time. A fourth limitation is that since all three locations observed were in relatively affluent communities in Houston, the individuals observed may not be representative of the entire population, and hence the external generalizability of the study may be affected. Since this was a serial cross-sectional study, the same people were not observed at each period. Despite these limitations, there was a standardized method of data collection. Another strength is that because this was an observational study, the data are more objective than the self-reported data. In addition, since the sample size for the observations was large, the results of the data analysis are robust.

## 5. Conclusions

Mask-wearing and social distancing behaviors of individuals at three outdoor locations in Houston, Texas were observed from 20 January to 30 April 2021. Data analysis from these observations indicates that these behaviors differed depending on gender, race, age, location, and when the mask mandate was and was not in place. The findings from this study can be used to inform the guidelines and policies regarding mask-wearing and social distancing behaviors in large urban city spaces. More specifically, policy-makers may want to consider differentiating between indoor and outdoor exposure to COVID-19 by emphasizing that while the risk of contracting COVID-19 outdoors is lower than the risk of contracting it indoors, it is not zero [22,23]. Hence, it is important to take the necessary precautions such as wearing a mask outdoors if individuals are not able to socially distance and are in crowded outdoor spaces such as at a concert.

## Figures and Tables

**Table 1 ijerph-20-02723-t001:** Descriptive statistics of the SOMAD observations in Houston, TX.

Demographic Characteristic	*n*	Percent of Total
**Race**		
White	4878	62.71
Latino	1275	16.39
Black	874	11.24
Asian	752	9.67
**Gender**		
Female	4154	53.41
Male	3624	46.59
**Age**		
Toddler (0–2)	38	0.49
Child (3–12)	480	6.17
Teen (13–19)	92	1.18
Adult (20–59)	6243	80.26
Senior (60+)	925	11.89
**Location**		
Farmer’s market	3311	42.57
Urban park	1878	24.15
Urban park with trail	2589	33.29
**Total**	**7778**	

**Table 2 ijerph-20-02723-t002:** Mask wearing behavior by statewide policy: Mask mandate vs. no mask mandate.

Location	Mask Mandate = Yes (% of *n*)	Mask Mandate = No (% of *n*)
Urban park	35.7	26.6
Urban park with trail	5.1	2.6
Farmer’s market	87.0	81.4

**Table 3 ijerph-20-02723-t003:** Logistic regression estimates predicting the mask-wearing behavior in response to COVID-19, Houston, TX, January to April 2021.

	Odds Ratio	95% CI
Male	1.00	
Female	1.35 ***	(1.18, 1.54)
White	1.00	
Latino	1.07	(0.89, 1.29)
Black	0.94	(0.76, 1.17)
Asian	1.84 ***	(1.48, 2.30)
Mask Mandate = No	1.00	
Mask Mandate = Yes	1.60 ***	(1.40, 1.84)
Adult (20–59)	1.00	
Toddler (0–2)	0.01 ***	(0.00, 0.10)
Child (3–12)	0.24 ***	(0.19, 0.31)
Teen (13–19)	1.54	(0.91, 2.61)
Senior (60+)	1.54 ***	(1.24, 1.91)
Farmer’s market	1.00	
Urban park	0.08 ***	(0.07, 0.09)
Urban park with trail	0.006 ***	(0.005, 0.008)

*** *p* < 0.001.

**Table 4 ijerph-20-02723-t004:** Logistic regression estimates predicting social distancing behavior in response to COVID-19, Houston, TX, January to April 2021.

	Odds Ratio	95% CI
Male	1.00	
Female	0.66 ***	(0.59, 0.73)
White	1.00	
Latino	0.75 ***	(0.65, 0.87)
Black	1.19 *	(1.01, 1.40)
Asian	0.68 ***	(0.56, 0.82)
Mask Mandate = No	1.00	
Mask Mandate = Yes	1.21 ***	(1.09, 1.34)
Adult (20–59)	1.00	
Child (3–12)	0.016 ***	(0.01, 0.04)
Teen (13–19)	0.23 ***	(0.12, 0.45)
Senior (60+)	1.29 ***	(1.11, 1.50)
Farmer’s Market	1.00	
Urban Park	0.88	(0.77, 1.01)
Urban Park with Trail	4.61 ***	(4.10, 5.17)

* *p* < 0.05; *** *p* < 0.001. Note: Toddlers were excluded from this model because they were always with adults and therefore were not socially distant from others.

## Data Availability

The data presented in this study are available upon request from the corresponding author.

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
