# Peer review of "Mask Adherence and Social Distancing in Houston, TX from January to April 2021"

_ijerph, 2023, doi:10.3390/ijerph20032723_

Round 1

Reviewer 1 Report

1) interesting resarch that may have an impact on public health activities, especially infrction prevention

2)large group but please supplement the provision on the obligation to wear masks - who and where, according to the order, had such an obligation - were the children also obliged to wear

2)pls increase/expand your references

Author Response

I am unable to find references that mention a mask mandate or any other order of similar nature regarding mask-wearing outdoors due to the lack of evidence that masks needed to be worn outside in a space where people were frequently moving. 

Reviewer 2 Report

This was an observational study in design to explore the differences of demographics, location, and social distance behaviors in response to mask mandate in Houston Texas.  The research is critical which could guide future decision-making processes for the implementation of effective intervention approaches in response to public health crises.  

There were several major flaws associated with study design that substantially dampened the significance of the study:

  1. The race/ethnicity of the participants were recorded through observing, therefore the potential of misclassification of race/ethnicity is high. The age misclassification could be an even more serious issue because the faces of certain races/ethnicities look much younger than faces of people of similar age group from other races (Yukio Shirakabe, etal.  A new paradigm for the aging Asian face, in Aesthetic Plast Surg,  2003 Sep-Oct;27(5):397-402.), let alone the faces were covered by the masks. Inaccurate information on race/ethnicity and age significantly impacts the results and even the direction of the associations in the regression analysis. 

  2. Due to the pandemic, most people chose not to or limit their visits to the public locations. Therefore, significant participant sampling bias existed. For example, it is likely healthy, young people or someone being vaccinated will be more likely to visit the public park during the mask mandate period. 

  3. Toddlers and children (3-12) were enrolled in the study; they were excluded from social distancing behavior analysis because apparently, they were with their parents/legal guardians or adult relatives in the public locations.  For the same token, toddlers, children or even teens should not be considered independent variables in the model to predict mask wearing behavior, because whether they were wearing masks or not is greatly influenced by the adults who accompanied them. 

  4. Due to the flawed design of the study, other key covariates such as economic status of the participants, their professions, as well as education level could not be collected, which significantly impacted the reliability of the derived associations.

  5. On line 122, the authors stated that “on average, 83.8% of people wore masks at the farmer’s market”. However, on line 194, it read: “high odds of mask use at the farmer’s market is not surprising because of signage at the front entrance that said, “no mask, no entry” and there were also several police officers that walked around the farmer’s market and observed people.” While not being able to reach 100% mask adherence in any scenario is understandable, still 83.8% seems low under the warning of “no mask, no entry” and in the presence of the police office.  Who were those not wearing masks at Farmer’s market? Were they toddlers, children or teens?  Such information should be available but was missing in the manuscript. 

  6. Individual’s vaccination status is an important factor affecting their mask adherence behavior. This information however was not discussed in the manuscript. Compared to adults, toddlers and children in this manuscript had lower odds of mask use. Although authors stated that toddlers and the young were exempted from wearing the masks, the availability of the size of masks that can fit toddlers and young children during the pandemic also affect the odds of mask adherence behavior in this age group.    

  7. What happened to Table 4, there were no results from table 4 presented in the results section. 

Minor comments:

There were a few typos that need to be corrected:

  1. Line 41: “Another study conducted by Khubchandani et al. found that women in older age 40

groups (≥ ?? ????? were more likely to wear masks than…”, half “)” was missing after the word “years”.

  1. aOR when first appeared in the manuscript needs to be defined: “Adjusted Odds Ratio” 

Author Response

  1. Yes, I agree and I have added these comments to the paper in the limitations section.
  2. I have also added this to the paper in the limitations section.
  3. Per one of the authors: " A limitation of the study is that each individual was recorded as a separate observation.  When the data were analyzed, we could tell whether each subject was distant from others, but there was no guarantee that nearby individuals were also recorded as observations.  Even if they were all recorded, there was no way to tell which subjects were seen together.   While it is likely that parents influenced their children with regard to mask use, there was no way to adjust for this given how all subjects were recorded as separate observations.   The data showed that practically no toddler or child subjects were socially distant.  We felt it was reasonable to conclude that they were accompanied by a parent or guardian since the observations were done in public places where the toddlers/children were unlikely to be left unattended."
  4. While that is correct, none of these variables were being studied to begin with as that was not the focus of this study. 
  5. While it is true that there was a sign saying "no mask, no entry" and there was police presence, mask wearing was not actually being enforced. Since the data collection took place about 2 years ago, I can not recall who was or was not wearing a mask at this time, nor was that the focus of this study. 
  6. Again, while an important and valid point, vaccination status was not a variable that was measured in this study nor was the size of masks. 
  7. I added in this information in the results section and fixed typos. 

Reviewer 3 Report

I think there is a missing piece that needs to be included, which is the relative risk of different settings be described in terms of recent literature on aerosol transmission being the most important means of infection. This allows the discussion to be supported with additional information that supports some decisions by individuals in terms of assessing relative risk. 

It's important to also include some conclusions in terms of policy recommendations, and not only a description of results. These results are useful in terms of designing risk communication strategies to the general population. We must all acknowledge that outdoor use of masks is appropriate in certain densities, but clearly not in the majority of outdoor settings, whereas indoor, close quarter mask use should be more the focus. All these things contribute to make this data one that allows focusing on what needs to be improved for future situations.

Author Response

I have added in a couple sentences addressing both your comment regarding differentiating between indoor and outdoor risk of contracting COVID-19 and your comment regarding policy recommendations. As I mentioned in the paper, there is limited research on the risk of COVID-19 transmission outdoors. But after hours of trying to find some concrete evidence, I used what I could find as evidence that COVID-19 transmission is lower outdoors than indoors. 

Round 2

Reviewer 2 Report

This is an observational study and the design of the study was flawed with many potential variables not being considered (please see my previous comments). I don't feel the revision provided by the authors significantly improved the quality of the manuscript.  

Author Response

Unfortunately, your recommendations cannot be accommodated at this time as we do not have the resources to re-conduct this study.